# Preparation and Properties of Polyvinylidene Fluoride Nanocomposited Membranes based on Poly(*N*-Isopropylacrylamide) Modified Graphene Oxide Nanosheets

**DOI:** 10.3390/polym11030473

**Published:** 2019-03-12

**Authors:** Xiangli Meng, Yuan Ji, Genhua Yu, Yujia Zhai

**Affiliations:** School of Chemistry and Chemical Engineering, Harbin Institute of Technology, Harbin 150001, China; JiYuan0713@163.com (Y.J.); 18846161758@163.com (G.Y.); zhaiyujia1992@126.com (Y.Z.)

**Keywords:** GO-g-PNIPAAm, ATRP, thermo-responsive, PVDF, ultrafiltration membranes

## Abstract

The nanomaterial of graphene oxide grafting poly (*N*-isopropylacrylamide) (GO-g-PNIPAAm) was synthesized and PVDF/GO-g-PNIPAAm blended membranes were fabricated by wet phase inversion. In this work, a hydrophilic nanomaterial GO-g-PNIPAAm with poly(*N*-isopropylacrylamide) (PNIPAAm) grafted on GO, was synthesized by the atom transfer radical polymerization (ATRP) method. The resulting nanomaterial was confirmed by scanning electron microscope (SEM), Fourier transform infrared spectroscopy (FTIR), Raman spectrum, and X-ray photoelectron spectroscopy (XPS) analysis. The synthesized GO-g-PNIPAAm was incorporated with polyvinylidene fluoride (PVDF) via phase inversion, and investigated for its temperature sensitivity, porosity, contact angle, scanning electron microscopy, and permeate properties. The water contact angle measurements confirmed that GO-g-PNIPAAm nanomaterial-endowed PVDF membranes with better hydrophilicity and thermo-responsive properties compared with those of the pristine PVDF membranes. Bovine serum albumin (BSA) adsorption experiments suggested that excellent antifouling properties of membranes were acquired after adding GO-g-PNIPAAm. The modified membranes showed good performance when the doping amount of GO-g-PNIPAAm was 0.2 wt %.

## 1. Introduction

Poly(vinylidene fluoride) (PVDF) has been widely used in separation such as water treatment [1,2] and bio-separation [3] due to its outstanding properties, like chemical resistance, thermal stability, high mechanical strength and good processability [4,5]. However, the serious protein-fouling problem had been the major drawback in the water treatment, because of its hydrophobicity [6]. Hydrophilicity is a vital property of ultrafiltration membranes in the application of water treatment. Nanocomposite membranes, blending inorganic particles with polymer materials, become an emerging technology for separation and purification in recent years. 

Nanoparticles including TiO_2_ [7,8], Al_2_O_3_ [9], SiO_2_ [10], graphene oxide [11,12,13], etc. have been reported excellent additives in hydrophilic modification, and the nanoparticles can also improve membrane structure, allowing the hybrid membrane to possess both high temperature resistance and well toughness [14].

Since most inorganic nanomaterials are insoluble, or they have a pronounced tendency to aggregate in water, the decoration of the NP surface with grafting polymer chains is a common strategy to obtain a good dispersion state of NPs in a polymer matrix [15]. Incorporating existing nanomaterials with multifunctional polymers allows them to not only retain their properties of interest, but also give access to novel fascinating structures and properties.

In recent years, much attention has been paid on stimulus-responsive membranes, which are sensitive to environmental stimuli such as temperature, pH, light, electric field [16,17,18,19], and so on. The temperature-sensitive ultrafiltration membranes can dramatically change their performance of permeability and separation when they suffer from surrounding temperature stimuli. They have attracted much of attention in the fields of controlled drug release [20,21], bio- or chemical separation [22,23], and water treatment [24]. 

PNIPAAm is the most common polymer, with thermal responsiveness that possesses both hydrophobic isopropyl and hydrophilic amidogen groups, and a lower critical solution temperature (LCST) is 32 °C in an aqueous solution [25]. Below the LCST, polymer chains have an extended random coil conformation in water. As the temperature is above the LCST, the polymer chains dehydrate to form a compact structure. Because of the excellent properties, PNIPAAm has been widely applied in thermo-responsive drug release, biosensors, bio-separation, anti-pollution self-cleaning membrane [26,27,28], and so on. 

To fabricate the temperature-sensitive membrane, a great variety of methods have been used, including plasma treatment, high energy radiation (UV, electron beam, γ-ray, etc.). However, the methods usually suffer permeability decreasing with pore-size distribution changing or property of original membrane materials changing by the grafted chains. Moreover, the grafting polymerization often occurs on the membrane surface, and it is difficult to reach the deep pore walls. In addition, the coverage of the grafted chains on the membrane surface may be not uniform [5].

To overcome the above-mentioned shortcomings, blending with nanoparticles grafting PNIPAAm may be a feasible approach to prepare well-defined temperature-sensitive polymeric filtration membranes. Zhou et al. [29] reported a PVDF nanocomposite membrane blended with PNIPAAm modified TiO_2_, the experiments indicated that the structure, hydrophilicity, separation performance, and mechanical properties of the membranes acquired great improvement. The well-dispersed TiO_2_-g-PNIPAAm endowed nanocomposite membranes with better thermo-responsive flux, lower bovine serum albumin (BSA) adsorption, a higher water recovery ratio, and a BSA rejection ratio compared with PVDF membranes. Cai et al. [23] reported a thermo-responsive PVDF/palygorskite-g-PNIPAAm hybrid ultrafiltration membrane prepared by blending the palygorskite-g-PNIPAAm nanofiber with PVDF, which exhibits temperature-responsiveness and enhanced antifouling ability.

GO has abundant functional groups on the surface, including hydroxyl groups, carboxyl groups, and epoxy groups, which make it possess strong hydrophilicity [30,31]. Meanwhile, the intrinsic properties of GO, such as good chemical stability, high surface area, and good mechanical properties, make it feasible as an additive for the fabrication of composite membranes [32,33] for the further functionalization of GO to provide active sites, relatively broadening its scope of application [34,35,36]. Recently, the chemical functionalization of GO by polymers and block copolymers has aroused great interest [37,38,39,40]. Chemically functionalized GO can tightly intertwine with the PVDF matrix due to their long polymer chains. As a result, the covalent functionalization not only makes the dispersity of GO better, but it also enables the interfacial interactions between graphene and matrix to be stronger [11,41].

Herein, a kind of inorganic–organic nanocomposite additive (GO-g-PNIPAAm) was synthesized via atom transfer radical polymerization (ATRP) using 2-chlorine propionyl chloride as the initiator. Compared to conventional polymerization methods, the ATRP process is applied in a variety of solvents, and it requires relatively mild reaction conditions. The polymer can be grown on the surface of materials including GO and nanoparticles, and the prepared polymer has a low molecular weight dispersibility index and an adjustable chain morphology and size, which is an effective way to carry out living polymerization. Then PVDF/GO-g-PNIPAAm-blended membranes were fabricated by wet phase inversion. In this system, PNIPAAm acts as the hydrophilic and thermo-responsive donor and the role of surface segregation, GO plays the role of both the hydrophilic modifier and the anchorage of PNIPAAm in the PVDF matrix. Hence, we can expect that the GO-g-PNIPAAm nanocomposite can improve the dispersity of GO NPs in the PVDF matrix, and the membrane antifouling performance. Besides, the effects of the PNIPAAm polymer chain on the GO surface will provide membranes thermo-responsive properties. The surface chemical composition, morphology, hydrophilicity, and temperature-sensitivity of the nanocomposite membranes were investigated in detail.

## 2. Materials and Methods

### 2.1. Materials 

PVDF (FR904) was purchased from Shanghai 3F New Materials Co. Ltd., Shanghai, China; GO was prepared in-house according to the modified Hummers’ method [42]; *N*,*N*’-dimethylformamide (AR, DMF) was received from Tianjin Fengchuan Chemical Reagent Science and Technology Company, Ltd, Tianjin, China. *N*-isopropylacrylamide (98%, NIPAAM) and triethylamine (AR,99%) were obtained from Aladdin Chemistry, Ltd., Shanghai, China. Tris(2-dimethylaminoethyl)amine,2-chloropropionyl chloride (95%), cuprous chloride (98%), 2-(2-aminoethoxy) ethanol, and *N*,*N*’-carbonyldiimidazole (98%) were ordered from Beijing Lark Technology Co., Ltd, Beijing, China; polyvinylpyrrolidone (PVP, K30), absolute ethyl alcohol (99.5%) and *N*,*N*’-dimethylacetamide (AR, DMAC) were provided by Tianjin Damao Chemical Reagent Factory; BSA was purchased from Beijing Biological Technology Co., Ltd., Beijing, China. These chemicals were used as received.

### 2.2. Synthesis of Amide Graphene Oxide(GO-1)

Dry GO (50 mg) was added into DMF solvent (20 mL), the mixture was sonicated for three hours so that it is sufficiently dispersed in the solution, followed by the introducing of catalyst *N*,*N*’-carbonyldiimidazole (30 mg). The resulting mixture was stirred at room temperature (25 °C) for 3 h, then added 2-(2-aminoethoxy) ethanol (1 mL) slowly (the dropping rate was 0.1 mL per minute) to the solution and continue stirred for 24 hr at room temperature. The resultant suspension was centrifuged, and washed several times using a solvent, then dried in a drying oven at 160 °C, the black solid marked as GO-1 was obtained.

### 2.3. Synthesis of Halogenated Initiators (GO-2)

Exactly 20 mg of GO-1 was added into 10 mL of toluene and sonicated for 1 hr to obtain a homogeneous colloidal suspension. Followed by introducing of triethylamine (1.5 mL) under the condition of ice bath, and the mixture was stirred for 2 h. Under the same conditions, 2-chloropropionyl chloride was added slowly (kept at 0.7 mL/min), continue stirred for 2 h. Then, the mixture was cooling reflux at 80 °C for 24 h. The resulting suspension was centrifuged and washed several times using toluene and deionized water, then dried in a drying oven at 110 °C, the black solid marked as GO-2 was obtained.

### 2.4. Synthesis of GO-g-PNIPAAM via ATRP Method

Three clean Schlenk bottles were prepared and marked as Sc1, Sc2, Sc3, respectively. Exactly 20 mg of GO-2 in DMF was added to Sc3 bottle, and the mixture was allowed to disperse uniformly for one hour. The PNIPAAm monomer (6 mmol) and DMF (2 mL) were added in Sc1 and the monomer was allowed to dissolve sufficiently, then added Me6TREN (0.24 mmol) with a micro-syringe. Cuprous chloride (24 mg, 0.24 mmol), DMF (2 mL) and deionized water (2 mL) were added into the Sc2 bottle. Then, nitrogen was bubbled into Sc1, Sc2, and Sc3, and the bottles were evacuated. The oxygen in the system was removed by freezing-evacuating-thawing, and circulated five times. The liquid in Sc3 bottle was thawed and injected into the Sc1 bottle with a double-headed needle, and the mixture was still sealed and stirred for 30 min. The GO-2 in the Sc2 bottle was then mixed with the first two by the same method, and the reaction was carried out for 24 hr under nitrogen protection at 60 °C. The resultant suspension was centrifuged and washed several times with DMF, then dried in a drying oven at 160 °C, the black solid marked as GO-g-PNIPAAm was obtained.

### 2.5. Preparation of PVDF/GO-g-PNIPAAm Nanocomposite Membrane

All the membranes were prepared by classical phase inversion method with PVDF as bulk material, DMAc as the solvent, PVP as pore-forming agent, GO-g-PNIPAAm as the additive according to the compositions listed in Table 1. The coagulation bath (distilled water) temperature was 30 °C. GO-g-PNIPAAm nanomaterials (0 wt %, 0.1 wt %, 0.2 wt %, 0.3 wt % and 0.4 wt % based on the weight of PVDF) and DMAc solvent were imported into a round-bottom flask, and then PVDF resin and PVP were added after the suspension was under ultrasonic dispersion for 2 hr. The casting solution was then mechanically stirred at 60 °C for 8 h until the polymer was completely dissolved. After fully degassing, the mixed homogeneous casting solution was spread onto a clean glass plate which was then immersed into a coagulation bath for 1 min. Subsequently, the resulting ultrafiltration membranes were rinsed in distilled water to remove the residual solvent, and dried at room temperature. The nanocomposite membranes were marked as M1, M2, M3, M4, and M5 according to the weight percentage of GO-g-PNIPAAm listed in Table 1. 

### 2.6. Characterization of Nanoparticles and Membranes

The chemical composition of nanoparticles and membranes were analyzed by Fourier transform infrared (FT-IR) spectroscopy (Nicolet’s Avatar 360 spectrum, Madison, WI, USA) and X-ray photoelectron spectroscopy (XPS, PHI 5700 ESCA System, American Institute of Physics, Chanhassen USA). Raman spectroscopy was taken on an inVia Spectra Raman system (London, UK).

Thermogravimetric analysis (TGA) was detected by the SDT Q600 instrument (Thermal Analysis Instruments, New Castle, DE, USA) at a heating rate of 10 °C·min^−1^ from room temperature to 800 °C under a nitrogen atmosphere. The thermo-sensitive properties of GO-g-PNIPAAm were also determined by differential scanning calorimeter (DSC) using the SDT Q600 instrument (Thermal Analysis Instruments, New Castle, DE, USA) with a temperature ranging from 25 to 50 °C at a rate of 2 °C min^−1^ in a nitrogen atmosphere.

GO-g-PNIPAAm and membrane surface morphology were observed using a scanning electron microscope (SEM, Quanta 200F, FEI Co. Ltd, Hillsboro, OR, USA).

### 2.7. Characterization of Ultrafiltration Membranes

Porosity *P*_r_ (%) refers to the number of holes per unit area, which indirectly reflects the size of the pores, but also the index of membrane permeability, which is calculated as follows:(1)$$P_{r}=\frac{\left(W_{w}-W_{d}\right)}{S\cdot d\cdot \rho }\times 100\%,$$ where *W*_w_ and *W*_d_ are the weight of the wet and dry film (g), *S* is the area of the measurement film (cm^2^), *d* is the average thickness of the film (mm) and ρ is the density of the distilled water at room temperature (g·cm^−3^). 

The hydrophilicity of membranes was evaluated by testing the contact angles (POWER2000 large-scale contact angle measuring instrument) of the dried membranes. At room temperature, a drop of deionized water was placed onto the surface of the ultrafiltration membrane sample on the stage. After 5 s, a photograph was taken, and the size of the water contact angle was calculated from the measurement accessory.

The water flux *J*_w_ (L·h^−1^·m^−2^) and the BSA rejection of the ultrafiltration membrane were tested by an MSC-300 cup ultrafilter; the effective test area of the ultrafiltration membrane was 36.32 cm^2^, nitrogen gas was utilized to force the water through the membrane, the film was preloaded at 0.2 MPa for 10 min before the test, and then it was measured at room temperature under a pressure of 0.1 MPa. The molecular weight of the BSA used was 68,000. The formula is as follows:(2)$$J_{w}=\frac{V}{S\times t},$$ where *J*_w_ is the permeation flux of the membrane for pure water (L·h^−1^·m^−2^), *V* is the permeate volume measured (L), *S* is the effective membrane area (m^2^), and *t* is the permeation time (hr).
(3)$$R=\left(1-\frac{C_{p}}{C_{f}}\right)\times 100\%,$$
where *R* is the rejection ratio (%); *C*_p_ is the concentration of the BSA permeate solution; and *C*_f_ is the initial concentration of BSA solution (g·L^−1^)

In the temperature sensitivity test for the membranes, the pure water flux and BSA rejection were tested at 25 and 40 °C, respectively. Also, the effects of different GO-g-PNIPAAm doping amounts on the blended membranes were investigated.

To investigate the antifouling properties of PVDF/GO-g-PNIPAAm-blended membranes, the flux recovery ratio (*FRR*) was calculated by using the following procedure: after pure water flux tests, the BSA solution was immediately replaced in the filtration cell and the flux of BSA solution was calculated based on the collected water weight at 0.1 Mpa for 30min. The fouled membranes underwent a distilled water batch for 30 min and the pure water flux was recorded again. Based on the obtained results, the *FRR* was calculated using Equation (4), and used to evaluate the antifouling performance of the membranes.

The fouling-resistant capacity of the membrane was further quantified with the total fouling ratio (*R*_t_), reversible fouling ratio (*R*_r_) and the irreversible fouling ratio (*R*_ir_), which were defined and calculated using the following expression, respectively:(4)$$\mathrm{FRR}=\frac{J_{w2}}{J_{w1}}\times 100\%,$$
(5)$$R_{r}=\frac{\left(J_{w2}-J_{P}\right)}{J_{w1}}\times 100\%,$$
(6)$$R_{\mathrm{ir}}=\frac{\left(J_{w1}-J_{w2}\right)}{J_{w1}}\times 100\%R,$$
(7)$$R_{t}=(1-\frac{J_{p}}{J_{W1}})\times 100\%R,$$
where *J*_w1_ is the water flux (L·h^−1^·m^−2^); *J*_w2_ is the water flux of the cleaned membrane (L·h^−1^·m^−2^), and *J*_p_ is the flux of the BSA solution (L·h^−1^·m^−2^).

## 3. Results and Discussion

### 3.1. Characterizations of GO-g-PNIPAAm

The FT-IR spectra of the PNIPAAM GO, GO-1, GO-2, and GO-g-PNIPAAm were presented in Figure 1. According to previous studies [36,43], where the characteristic peaks of the original GO are at 3380, 1722, and 1610 cm^−1^ for the –OH stretching vibrations, >C=O stretching vibrations in carboxyl, C=C stretching vibrations of the unoxidized graphitic domains, respectively. The peaks at 1215 and 1042 cm^−1^ are both for C–O stretching vibrations of epoxy functional group. For GO-1, the absorption peak at 3368 cm^−1^ corresponds to the N–H bonds and peaks at 2924 cm^−1^ and 1646 cm^−1^ correspond to the –CH_2_ and C=O stretch, respectively. The appearance of –CH_2_, N–H and the disappearance of the carboxyl peak demonstrate the occurrence of GO amination reaction. The FTIR spectrum of the GO-2 is distinguished by the strong absorption band at 3424 cm^−1^, as the −OH was consumed, the −OH broad peak disappears and the N–H stretching vibrations became obviously new peak at 1368 cm^−1^ is the deformation vibration of –CH_3_, 1567 cm^−1^ is C–N band in amide group and 1721 cm^−1^ owns to C=O in the ester group, the presence of which indicates the synthesis of a halogenated initiator. The appearance of two peaks at 1384 and 1366 cm^−1^ for GO-g-PNIPAAm corresponds to the characteristic bands of PNIPAAm, the results demonstrate that PNIPAAm has been successfully grafted on the GO. Nevertheless, another interesting thing is that the peak at 3300 cm^−1^ is significantly wider than that of GO-2, possibly due to the superposition of free-state and the hydrogen bond association of N–H stretching vibration peaks. These results will be further provided below.

Raman spectroscopy is used to identify the crystalline structures of ordered and disordered carbon materials. The Raman spectrum of GO (Figure 2) exhibits two obvious bands at 1343 cm^−1^ (D band) and 1596 cm^−1^ (G band) arising from the vibration of *sp*^3^ carbon atoms from the functional groups, reflecting the defects of the internal structure of GO and the in-plane vibration of *sp*^2^ carbon atoms, which indicate the GO structure of the symmetry and order, respectively. In Raman analysis, the intensity ratio of the D peak to the G peak (*I*_D_/*I*_G_) was used to denote the disorder degree of GO [43]. The higher the ratio, the greater the disorder degree and the more defects. It was found that the *I*_D_/*I*_G_ ratio of GO was 0.82. Compared with GO, the *I*_D_/*I*_G_ ratio of GO-1 increases to 0.89. The higher *I*_D_/*I*_G_ ratio of GO-1 gives an indication of the formation of comparatively smaller *sp*^2^ domain size than GO, suggesting that some of the ordered structure of GO were further destroyed by the grafting of 2-(2-aminoethoxy) ethanol. With the progress of the grafting reaction, the intensities of the D and G peaks decreased significantly and widened, indicating that the ordered structure of GO was destroyed and the transition from *sp*^2^ to *sp*^3^ was observed with the grafting reaction. In addition, it is also stated that the grafts on GO gradually increase, and their own active sites are covered, resulting in a decrease in the strength of the two peaks. Especially for the grafted GO-g-PNIPAAm, the peak intensity reduction is extremely obvious, the ratio of *I*_D_ / *I*_G_ increased to 0.95.

In order to further verify the PNIPAAm contents on the GO surface in detail, the X-ray photoelectron spectroscopy (XPS) was tested. Figure 3a shows the XPS spectra with a wide scan of GO, GO-2, and GO-g-PNIPAAm, and the elements content analysis is summarized in Table 2. As seen in Figure 3a and Table 2, only C and O bands were observed for GO, and with the graft reaction, the content of N element gradually increased. GO-g-PNIPAAm shows the highest content of N elements, indicating that PNIPAAm was grafted onto GO surface. As seen in Figure 3b, the C1s peaks of GO can be deconvoluted into four components at about 284.6, 285.7, 286.6, and 288eV, corresponding to carbon atoms in different functional groups: C–C, C–OH, C–O–C, and O–C=O [33]. For GO-2, apparent new peaks of N1s and Cl 2p were obtained besides C and O elements in Figure 3a. Figure 3c illustrates that there are two characteristic peaks at 399.5 eV and 398.7 eV in the N1s spectrum, corresponding to CONH_2_ and C–N, which are the powerful evidence of the successful functionalization of GO by 2-(2-aminoethoxy) ethanol. For GO-g-PNIPAAm, the elements obtained are C, O, and N. Figure 3d reveals that the contents of O and C were drastically reduced while the content of N was significantly increased, and the content of Cl was almost zero, which indicated the grafting of PNIPAAm onto the surface of GO successfully. This result was consistent with the observation from FT-IR spectra and Raman spectroscopy. As can be seen in Table 2, for GO-g-PNIPAAm, the C 1s content decreases from 76.42% (GO-2) to 72.56%, besides, the N1s content changes from 2.17% (GO-2) to 7.13%, due to the introduction of amide groups of PNIPAAm. The content of Cl2p element is changed from 2.11% (GO-2) to 0.23% (GO-g-PNIPAAm). All of the results demonstrate the successful grafting of PNIPAAm onto the GO surface.

TGA was performed to study the decomposition behavior and the thermal stability of GO and the GO-g-PNIPAAm composite material. As shown in Figure 4, from the curve of GO, the loss of adsorbed water in its π stack structure is about 10% from the beginning of heating to 150 °C [38]. However, a relatively large weight loss of about 25% occurs from 150 °C to 400 °C, due to decomposition of oxygen-containing groups [39]. It can be seen from the thermogravimetric curve of GO-g-PNIPAAm that the decomposition temperature range is between 180 °C and 550 °C, higher than that of GO. Similar conclusions on the graphene grafted PMMA, the thermal stability of the nanocomposite particles increased [41]. From the TGA curve, we could see that the grafting ratio is about 14%.

The SEM images of GO, GO-1, GO-2, and GO-g-PNIPAAm are shown in Figure 5. As can be seen from Figure 5, the layered GO surface is relatively smooth, and the edge is also regular. While for GO-1 and GO-2, the edge becomes gradually wrinkled, and the surface becomes rough. After grafting the copolymer, the folded morphologies and the thickness of the sheet became larger. In addition, we can clearly see that polymeric chains of PNIPAAm covered the surface of GO, and the change in the thickness and edge morphology of GO illustrated the successful grafting of PNIPAAm onto the GO surface.

The thermo-sensitive properties of GO-g-PNIPAAm were determined by DSC, with temperatures ranging from 25 °C to 50 °C, which is shown Figure 6. Compared with the curve of the original GO, there is a maximum endothermic peak around 32 °C for GO-g-PNIPAAm, which is consistent with the phase transition temperature (LCST) of PNIPAAm. This means that GO-g-PNIPAAm owns thermo-sensitivity properties, as well [11].

### 3.2. Characterizations of Membrane Structures

In order to investigate the membrane morphologies affected by GO-g-PNIPAAm, the surface and cross-section morphologies of the pristine membrane (M1) and the blended membranes (M2~M5, doping amount of GO-g-PNIPAAm was 0.1 wt %, 0.2 wt %, 0.3 wt % and 0.4 wt %, respectively) were studied via SEM, and the results are given in Figure 7. From the top surface, it is obvious that the number of microvoids increased and then decreased with GO-g-PNIPAAm content increased. When the doping amount of GO-g-PNIPAAm is 0.2 wt %, the membrane has the largest pores due to the immediate demixing formed by filler nanomaterials during the phase inversion process. And, a better hydrophilicity of GO-g-PNIPAAm would promote better solvent diffusion from the polymer matrix to water, which would facilitate the formation of a larger pore density. When doping amount of GO-g-PNIPAAm is more than the 0.2 wt %, pore size decreased. That may be because the addition of GO-g-PNIPAAm may increase the viscosity of dope solution, which retarded the solvent exchange rate and delayed demixing, in favor of the decrease in pore size. The same trend can be seen from the cross-sectional images. With the doping amount increase, the size of finger holes gradually become thicker. When the doping amount was 0.2 wt %, the membrane pore structure was the best, showing the even finger-like pore structure. When the doping amount was over 0.3 wt %, as shown in Figure 7(d’),7(e’), the transverse pore structure appears in the film, which may be due to the fact that more GO-g-PNIPAAm will be laterally distributed in the casting solution or accumulate, and the presence of transverse pores interrupts the connected finger holes, which may affect the permeability of the blended membrane and reduce the water flux.

To further investigate the impact of membrane structure on permeability, the porosity is described in Figure 8. The porosity increased with the increasing amount of GO-g-PNIPAAm in casting solutions. The porosity acquired the maximum at 0.2 wt % of GO-g-PNIPAAm. The trend also can be clearly observed in SEM images (Figure 7). The results demonstrated that the porosity could be controlled by different doping amount of GO-g-PNIPAAm.

### 3.3. Surface Analysis of the PVDF/GO-g-PNIPAAm Membrane by XPS

The surface compositions of membranes are studied by XPS. Figure 9 shows the wide-scan spectra for PVDF and PVDF/GO-g-PNIPAAm, respectively. As seen in Figure 9, for PVDF, only the C and F bands were observed at 290.8 eV and 694.0 eV, respectively. As for PVDF/GO-g-PNIPAAm blended membrane, two new peaks were observed, 406 eV for N1s and 537.2 eV for O1s. It is indicated that the GO-g-PNIPAAm nanocomposite was blended into the PVDF matrix, and migrated to the membrane surface successfully.

### 3.4. Membrane Hydrophilicity Analysis

The water contact angle is often used to evaluate the hydrophilicity of the membranes. Figure 10 shows the dynamic water contact angle measured at room temperature (25 °C) every 20 s and the test time is 120 s. As shown in the figure, the contact angle of all the membranes exhibit an attenuation. Moreover, it is obvious to see that the initial contact angle dropped as the doping amount of GO-g-PNIPAAm increased. These results are ascribed to the incorporation of amphiphilic additives in the composite membranes. As is well-known, when the water contact angles were tested at 25 °C, the PNIPAAm had a hydrophilic extended conformation, and the water drop was absorbed rapidly. As illustrated in Figure 10, it is obvious to see that the initial contact angle of M3 is the lowest (75°), approximately 10° lower than the pure PVDF membrane (M1), and after 120 s, the value dropped about 15°, while only dropped 7° for the pristine one, indicating the enhanced hydrophilicity and permeability ascribed to the incorporation of GO-g-PNIPAAm. Interestingly, when the doping amount of GO-g-PNIPAAm is more than 0.2 wt %, the initial contact angle increased. This phenomenon can be analyzed from below: as PNIPAAm is hydrophilic, so that the membrane hydrophilicity increased with the increase of GO-g-PNIPAAm on the membrane surface. While the contact angle increased with the addition of GO-g-PNIPAAm over 0.2 wt %, the viscosity of the dope solution increased, which retarded the solvent exchange rate during the phase-inversion process. Consequently, the GO-g-PNIPAAm nanomaterial cannot move to the surface of the membrane immediately, resulting in an increase of contact angle.

### 3.5. Permeability of Membranes

In order to investigate the filtration performance of PVDF/ GO-g-PNIPAAm membranes, a pure water flux (PWF) and BSA rejection ratio at 25 °C were measured. The effect of the weight fraction of GO-g-PNIPAAm is shown in Figure 11. As shown in Figure 11, the water flux of pure PVDF membrane is only 496 (L·h^−1^·m^−2^), with the increase of GO-g-PNIPAAm content, the water flux of the composite membranes increased gradually. When 0.3 wt % of GO-g-PNIPAAm was added in membrane, the water flux reached the maximum value of 826 (L·h^−1^·m^−2^). Subsequently, the water flux is slightly reduced. This may be due to the large doping amount of GO-g-PNIPAAm, while the BSA rejection ratio reached the minimum value, but it still reached more than 87%. This indicates that the proper addition of GO-g-PNIPAAm is attributed to the enhancement of membrane hydrophilicity, as well as the variation of membrane structure.

Water flux was measured at 25 and 40 °C to investigate the temperature-sensitivity of water permeation properties for the PVDF and PVDF/ GO-g-PNIPAAm membranes. The results are shown in Figure 12. It was observed that the permeate flux of all of the membranes at 40 °C were obviously much higher than at 25 °C. This phenomenon can be analyzed from the following perspectives: firstly, the viscosity of pure water decreased at higher temperature. On the other hand, as described in the introduction, PNIPAAm is hydrophilic and soluble below its LCST (32 °C), and when the temperature is higher than the LCST, it becomes hydrophobic, and it shrinks in an aqueous solution. At 25 °C, the PNIPAAm polymer side chains had an extended conformation; as a result, the effective pore size of membrane was narrowed, which led to a lower water flux. As the temperature raised to 40 °C, the PNIPAAm chains became hydrophobic and formed a compact structure, leading to a release of pore size, therefore showing in increased water flux.

The temperature sensitivity of the modified membranes was estimated by comparing the values of *J*_40_/*J*_25_. When GO-g-PNIPAAm is added into the membrane, the *J*_40_/*J*_25_ value are larger than that of the pure PVDF membrane, a maximum of 1.79 occurs when the doping amount is 0.2 wt %. When the doping amount is more than 0.2 wt %, the *J*_40_/*J*_25_ value gradually becomes smaller, which may be due to the small porosity. The results indicated that the flux of membranes was decided by LCST and pore parameters. The hydrophilic/hydrophobic transition of PNIPAAm polymer brushes occurred around the LCST led to the changes in flux, which resulted in the thermo-responsive performance of membrane pores [29].

The BAS rejection ratio of membranes at different temperature was illustrated in Figure 13. It can be found that the rejection ratio of blended membranes obtained a decline, compared to that of the PVDF membrane, and the effect at 25 °C was better than that at 40 °C. This is because the PNIPAAm brushes huddled together causing the “open” state of pores, results in the declined rejection ratio. Moreover, the nanocomposite membranes exhibited a decreasing BSA rejection ratio, along with the increase of GO-g-PNIPAAm, and the rejection ratio reached the minimum when the doping amount was 0.3 wt %. This corresponds exactly to the maximum value of the water flux (Figure 11).

To further investigate the fouling resistance of membranes in detail, dynamic protein filtration was performed. The *J*_w1_, *J*_p_, *J*_w2_ and water recovery ratio of membranes at 25 °C are shown in Figure 14. The flux recovery ratio (*FRR*) of the bare PVDF was only 47.6%, which was much lower than that of membranes that are modified by GO-g-PNIPAAm, indicating that the modified membranes can be cleaned easily by water. The reason is that when the temperature is below LCST of PNIPAAm, the hydrophilic GO-g-PNIPAAm on the membrane surface can form a layer of water molecules to avoid the bovine serum protein on the membrane surface deposition. The *FRR* of all blended membranes was more than 51.5%. The maximum value of *FRR* was 83.3% and consequently the best antifouling performance was observed for 0.2 wt % of GO-g-PNIPAAm. When the doping amount is more than 0.2 wt %, the viscosity of dope solution increased, which retarded the solvent exchange rate and delayed demixing, more effective functional groups were covered by polymer, which decreased the hydrophilicity of membrane surface; thus the antifouling performance reduced.

Membranes fouling can be clearly characterized by total fouling ratio (*R*_t_), reversible fouling ratio (*R*_r_) and irreversible fouling ratio (*R*_ir_). In reversible fouling, the flux decline could be reclaimed by water washing, because of the loose attachment of the foulants on the membrane. This type of fouling decreases the membrane productivity and increases the operational costs [44]. While in irreversible fouling, the flux could not be recovered by hydraulic washing because the foulants are strongly attached to the membrane, and chemical cleaning is needed to remove them. The calculated values of *R*_t_, *R*_r_, and *R*_ir_ are presented in Figure 15. As can be seen, the bare PVDF membrane has the highest *R*_t_ (83.3%) and *R*_ir_ (52.4%), while in the blended membranes, the *R*_t_ and *R*_ir_ was reduced. In the case of the blended membranes, the hydrophobic adsorption of BSA protein on the membrane were easily removed during filtration. The results also showed that the PVDF membranes contained 0.2 wt % of GO-g-PNIPAAm had great antifouling ability. It has the lowest *R*_t_ (72%) and *R*_ir_ (16.7%), the highest *R*_r_ (55.4%).

## 4. Conclusions

In summary, the GO-g-PNIPAAm nanocomposite synthesized by using the ATRP method was introduced into the basement membrane material to fabricate the PVDF/GO-g-PNIPAAm nanocomposite membranes for water treatment. As a surface additive, GO-g-PNIPAAm can migrate to membrane surface via segregation, and can be well-dispersed in the as-prepared PVDF/GO-g-PNIPAAm nanocomposite membranes. The contact angle measurements showed that the modified membranes possessed enhanced hydrophilicity. The permeability results showed that the PVDF/GO-g-PNIPAAm nanocomposite membranes had typical temperature-sensitive behaviors; that is, a sharp water flux change appeared at 40 °C. The well-dispersed GO-g-PNIPAAm endowed nanocomposite membranes with better thermo-responsive flux, higher water recovery ratio, BSA rejection ratio, and anti-fouling performance compared with those of PVDF membranes. The improvements should be attributed to the GO-g-PNIPAAm in nanocomposite membranes. Based on the comprehensive analysis of the test results, it is concluded that the best addition of GO-g-PNIPAAm in the ultrafiltration membrane is 0.2 wt %.

## Figures and Tables

**Figure 1 polymers-11-00473-f001:**
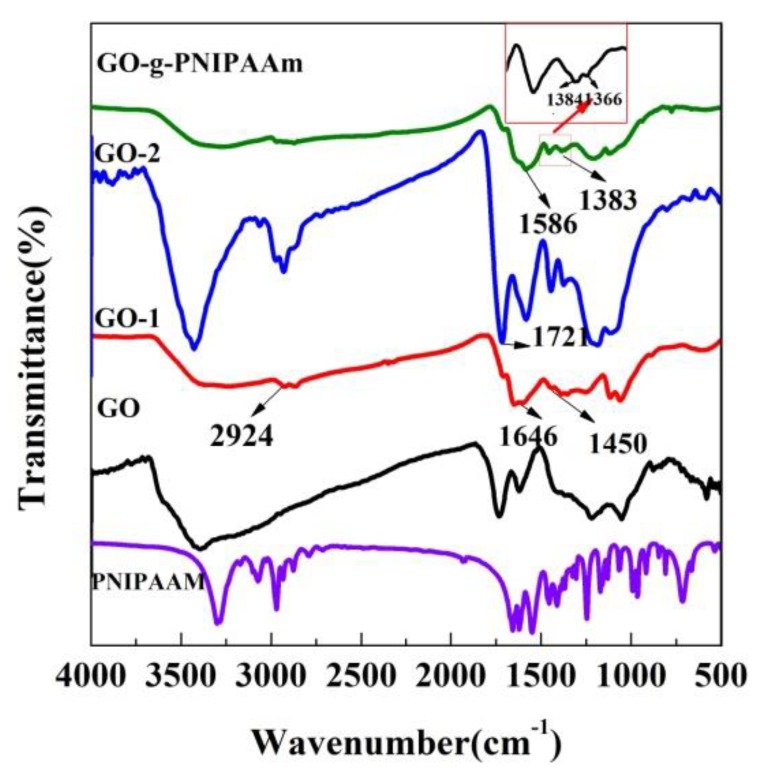
FT-IR spectra of the PNIPAAm GO, GO-1, GO-2 and GO-g-PNIPAAM.

**Figure 2 polymers-11-00473-f002:**
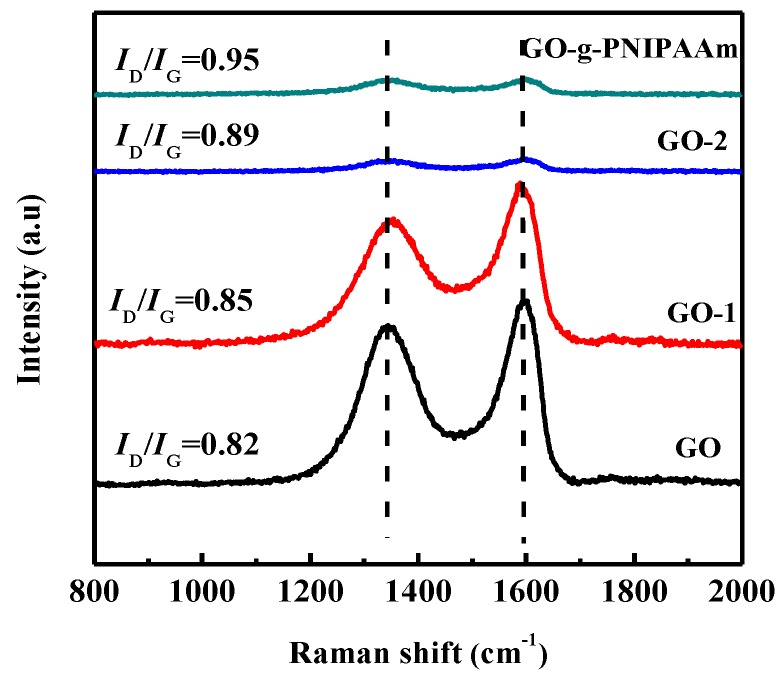
Raman spectrum of GO, GO-1, GO-2 and GO-g-PNIPAAm.

**Figure 3 polymers-11-00473-f003:**
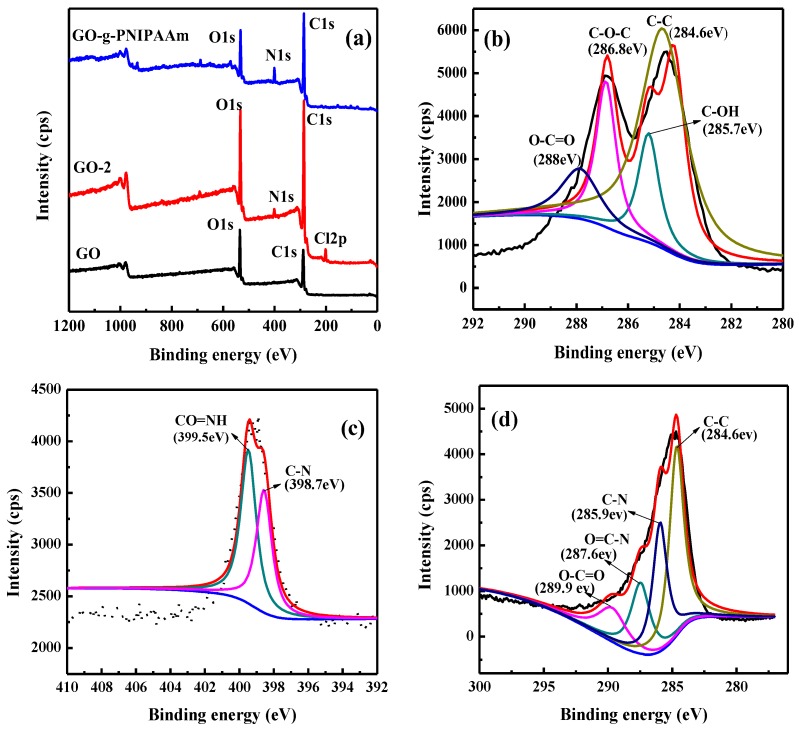
XPS-wide scan spectra of GO, GO-2, and GO-g-PNIPAAm (**a**), C1s spectra of GO (**b**), N1s spectra of GO-2 (**c**), and the C1s spectra of GO-g-PNIPAAm (**d**).

**Figure 4 polymers-11-00473-f004:**
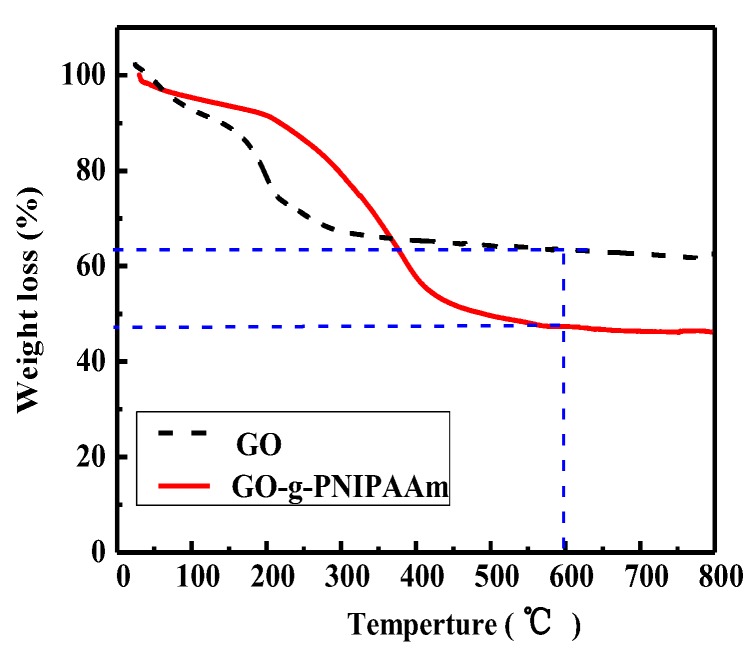
TGA thermograms of GO and GO-g-PNIPAAm in the N_2_ atmosphere.

**Figure 5 polymers-11-00473-f005:**
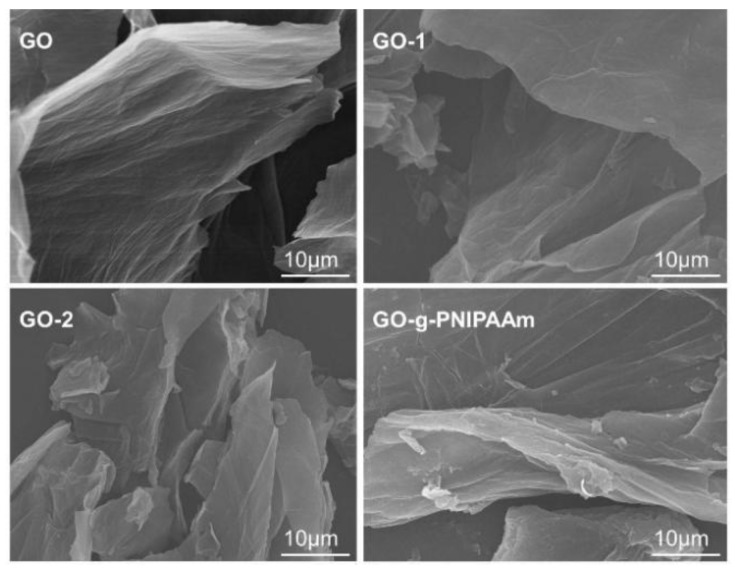
SEM images of GO, GO-1, GO-2, and GO-g-PNIPAAm

**Figure 6 polymers-11-00473-f006:**
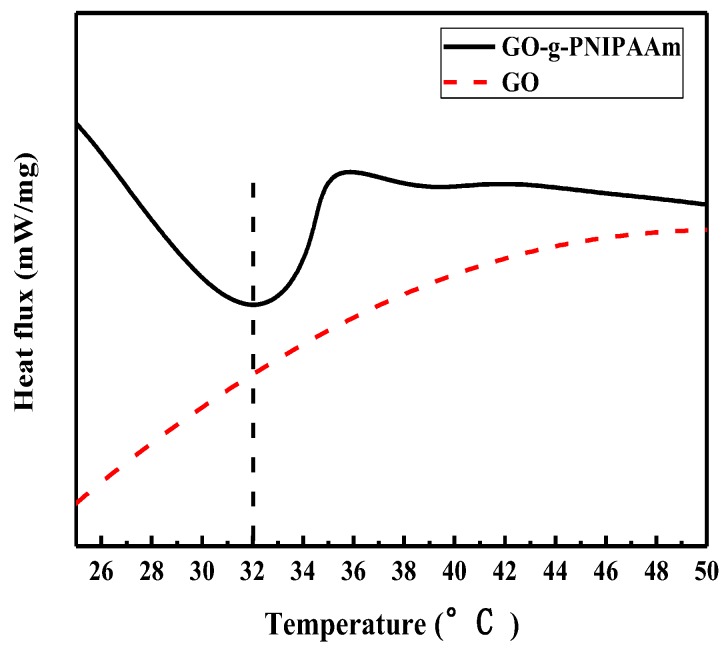
DSC curves of GO and GO-g-PNIPAAm.

**Figure 7 polymers-11-00473-f007:**
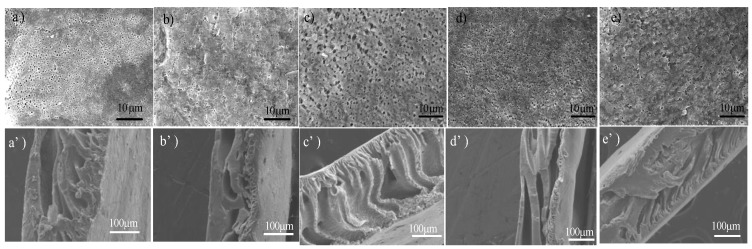
SEM images of membranes: (**a**)–(**e**) surface morphologies of M1~M5, (**a’**)–(**e’**) cross-sectional morphologies of M1~M5.

**Figure 8 polymers-11-00473-f008:**
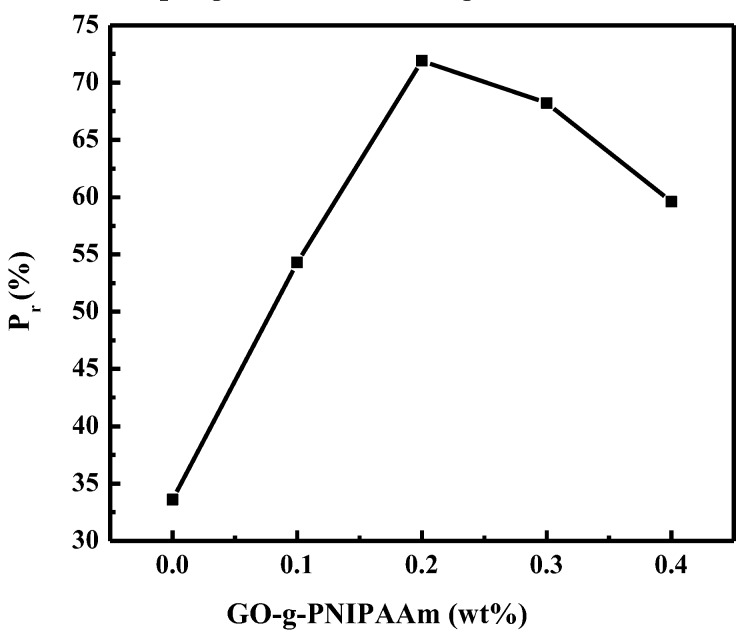
Porosity of membranes under different doping amount.

**Figure 9 polymers-11-00473-f009:**
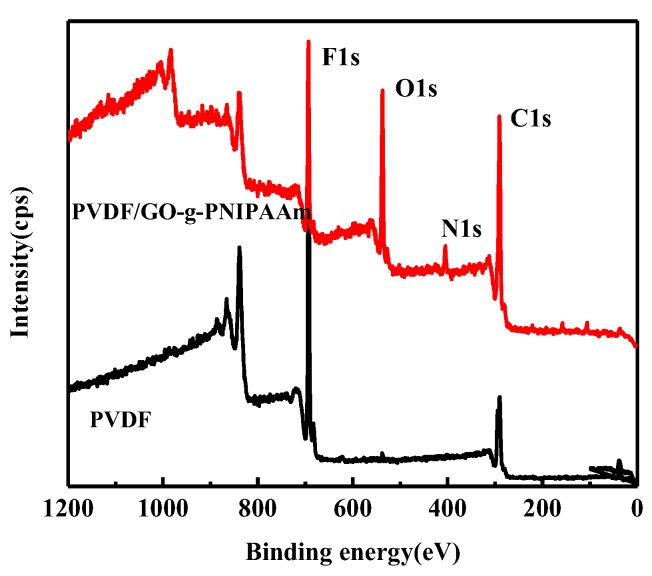
XPS-wide spectra of PVDF, and the typical PVDF/GO-g-PNIPAAm blend membranes.

**Figure 10 polymers-11-00473-f010:**
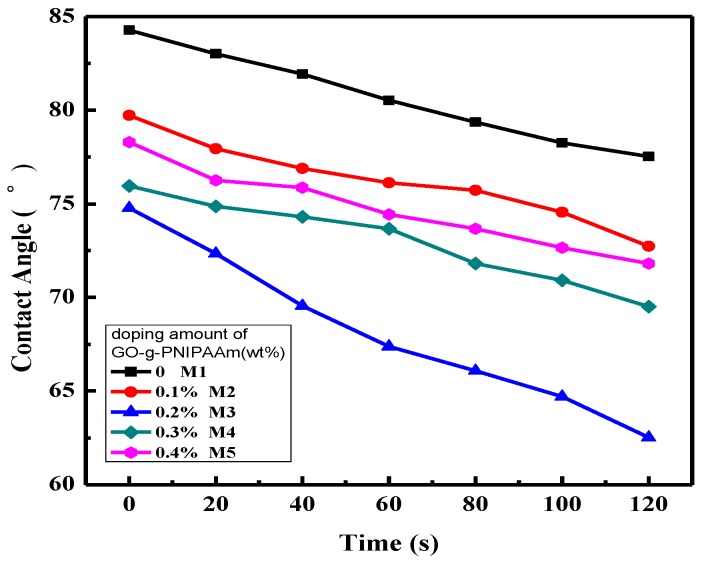
The water contact angle of membranes of PVDF and PVDF/GO-g-PNIPAAm composite membranes.

**Figure 11 polymers-11-00473-f011:**
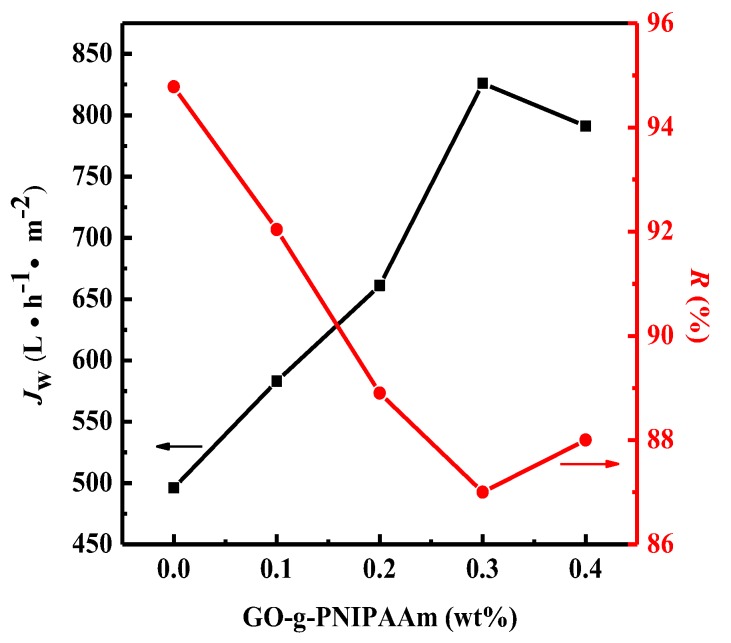
Water flux and BSA rejection of PVDF/GO-g-PNIPAAm membranes

**Figure 12 polymers-11-00473-f012:**
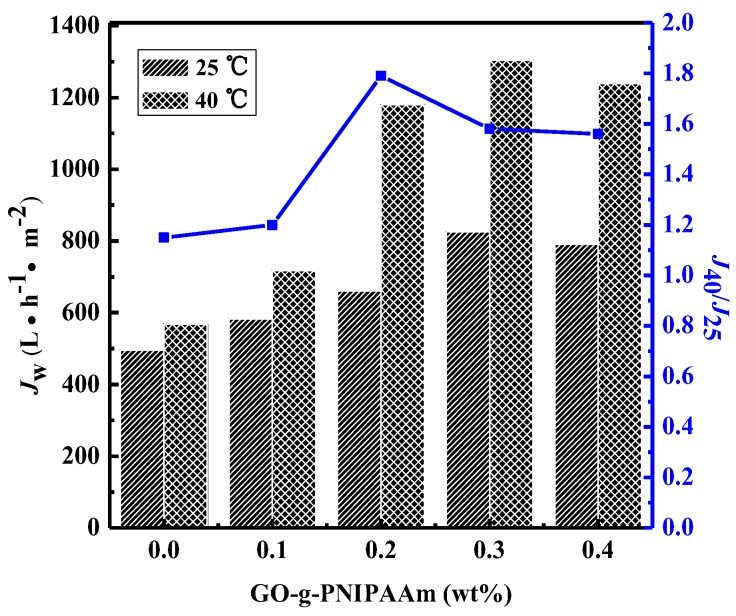
Water flux of the nanocomposite membranes at 25°C and 40°C.

**Figure 13 polymers-11-00473-f013:**
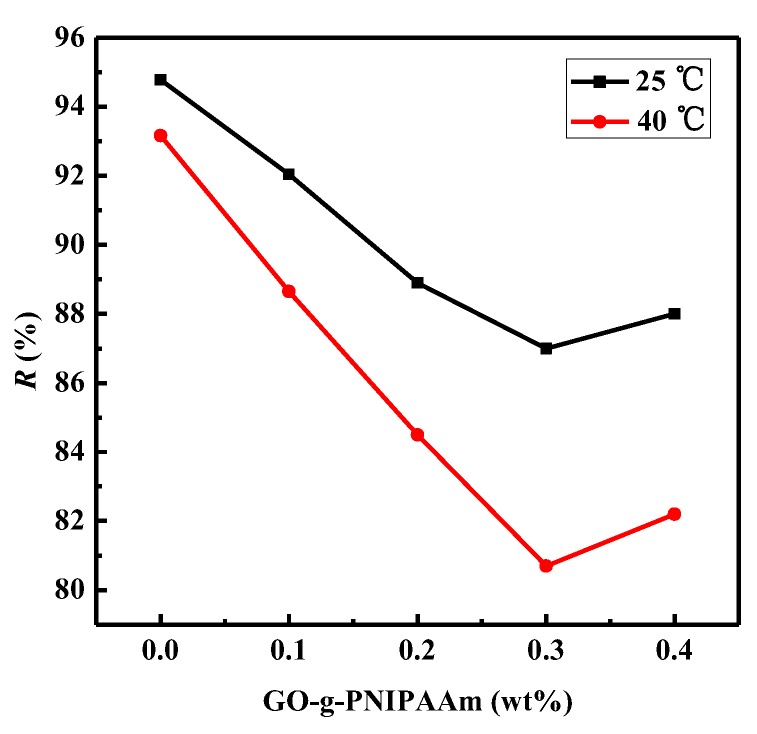
BSA rejection ratio of membranes at 25 °C and 40 °C

**Figure 14 polymers-11-00473-f014:**
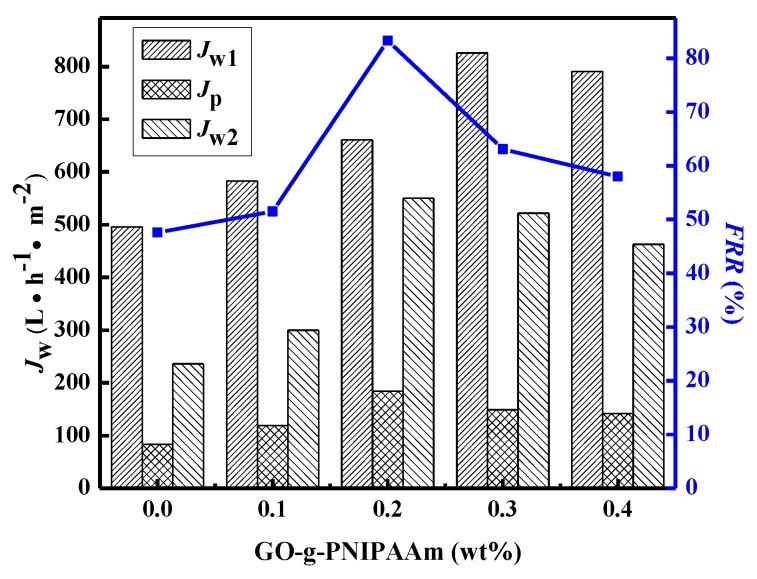
Permeation fluxes of water and BSA solution, and the water recovery ratio of one cycle for the membranes at 25 °C.

**Figure 15 polymers-11-00473-f015:**
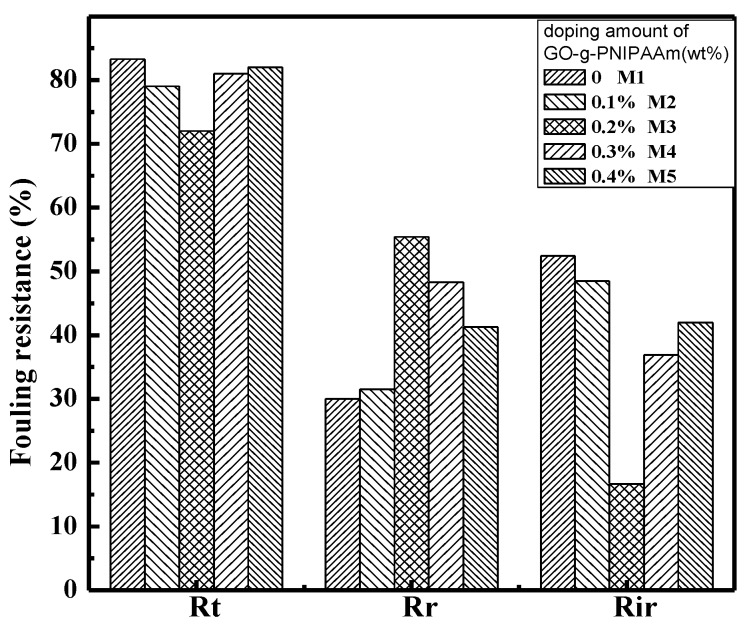
Fouling resistance ratio of the prepared PVDF membranes.

**Table 1 polymers-11-00473-t001:** Preparation of temperature-sensitive ultrafiltration membrane design table.

Membranes	GO-g-PNIPAAm (wt %)	PVDF (wt %)	PVP (wt %)	DMAC (wt %)
M1	0	20	3	77.0
M2	0.1	20	3	76.9
M3	0.2	20	3	76.8
M4	0.3	20	3	76.7
M5	0.4	20	3	76.6

**Table 2 polymers-11-00473-t002:** Element content analysis of GO, GO-2, and GO-g-PNIPAAm.

	Element	C1s (%)	O1s (%)	N1s (%)	Cl2p (%)
Samples					
GO		73.18	26.10	0.64	0.07
GO-2		76.42	19.30	2.17	2.11
GO-g-PNIPAAm		72.56	16.17	7.13	0.23

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
