# Peer review of "Preparation and Properties of Polyvinylidene Fluoride Nanocomposited Membranes based on Poly(N-Isopropylacrylamide) Modified Graphene Oxide Nanosheets"

_polymers, 2019, doi:10.3390/polym11030473_

Round 1

Reviewer 1 Report

The paper describes the synthesis and characterization of temperature sensitive membranes. In my opinion all the experiments are well conducted and the conclusions are supported by the results. I agree this paper to be published .

Author Response

Response to Reviewer 1 Comments

No questions or comments need to be response.

Reviewer 2 Report

The manuscript presents a novel composite-PVDF membrane presenting higher hidrophilicity and lower protein fouling effect for water filtration. Though the material reported is new, I have the following concerns:

Introduction: in general, english revision should be done, as the meaning of some sentences is not clear.

Materials and methods section:

a) what is the final ratio GO/PNIPAAm present in the composite?

b) What is the coagulation bath used?

b) In table 1 the total percentage for M2-M5 is higher tan 100%. As the GO-g-PINPAAm content increases some of the other components, usually the solvent, decreases to adjust to the 100wt%

c) Ecuations 5 and 6 are the same. The definition of irreversible fouling ratio RiR is missing, please, include it

Results and discussion section:

a) FT-IR of the PNIPAAm could be included to compare with the typical bonds at 1384 and 1366 cm-1 indicated in the text?  could the spectra be normalized somehow to omit the effect of the concentration in each material on the intensity of the bonds?

b) The BSA rejection decreases with composite content. This is a negative effect on UF filtration membranes. To what extent the BSA rejection decrease can be assumed for the real application of these composite-membranes?

c) The comparison of the FRR and fouling parameters at 25 and 40ºC should be included, as the thermo-responsive propertie of the membranes is the main novelty claimed in this manuscript

Conclusions section: as mentioned above, the major claiming of the novel membranes is the advantage of thermo-responsive effect on the water flux properties. As BSA rejection decreases so importantly when doping the PVDF membrane with the nanocomposite, I still have my concerns on the potential benefit of these membranes for this application, and more importantly, the advantage that the thermo-responsive effect incorporates to the real application.

Overall, I consider that this manuscript should have major revisions before acceptance.

Author Response

Point 1: what is the final ratio GO/PNIPAAm present in the composite?

Response 1:In this experiment, 0.2wt% was selected as the doping amount of GO-PNIPAAm in the composite.

Point 2: What is the coagulation bath used?

Response 2:In the experiment, we used distilled water as the coagulation bath and it was added to the article

Point 3:In table 1 the total percentage for M2-M5 is higher tan 100%. As the GO-g-PINPAAm content increases some of the other components, usually the solvent, decreases to adjust to the 100wt%

Response 3: This part of the article has been corrected in accordance with your request.

Point 4:Ecuations 5 and 6 are the same. The definition of irreversible fouling ratio RiR is missing, please, include it

Response 4:This part of the article has been corrected in accordance with your request.

Point 5:FT-IR of the PNIPAAm could be included to compare with the typical bonds at 1384 and 1366 cm-1 indicated in the text?  could the spectra be normalized somehow to omit the effect of the concentration in each material on the intensity of the bonds?

Response 5:According to your request, the FT-IR spectrum of PINPAAm has been added to the FT-IR spectrum part of the paper.

Point 6:The BSA rejection decreases with composite content. This is a negative effect on UF filtration membranes. To what extent the BSA rejection decrease can be assumed for the real application of these composite-membranes?

Response 6:In this paper, the experimental conditions of GO-g-PNIPAAm doping amount of 0.2wt% are selected, and the BSA rejection of UF membrane rate can reach 87%. According to the previous literature, this result is in line with the practical application conditions.

Point 7:The comparison of the FRR and fouling parameters at 25 and 40ºC should be included, as the thermo-responsive propertie of the membranes is the main novelty claimed in this manuscript

Response 7:In the experiment, we only made the FRR of the film at 25 °C to reflect the antifouling performance of the film. If data supplement is needed, we may need more time.

Reviewer 3 Report

The manuscript "Temperature sensitive Polyvinylidene Fluoride Nanocomposited Membranes based on  Poly(N-isopropylacrylamide) Modified graphene oxide Nanosheets"  presents the formation of hydrophilic temperature-sensitive nanomaterial GO-g-PNIPAAm.

So far there some issues to address. If you use GO as abreviation in abstract please explain what that is? The other part which is not clear written out what is the novelty and goal of this work? please define a clear goal in abstract

There far too many Figures shown. Combine some together or show some in supplementary. A scientific paper needs to show precise results and Figures with a proper discussion. There is no proper discussion with references to other work presented here. As example if you show FTIR or Raman with signals you need to give the origin of the lines (references). This has to be included as well for other experimental data.

There are no error bars shown in your experiemtal data. Did you investigate only one sample? Please include error bars and explain where they come from.

The addition of polymers on graphene oxide over click chemistry is well known and there many reviews and papers written about this topic

As example:

Y. Pan, H. Bao,N.G. Sahoo, T.Wu, L. Li,Water-soluble poly(N-isopropylacrylamide)–
graphene sheets synthesized via click chemistry for drug delivery, Adv. Funct.
Mater. 21 (2011) 2754–2763.

The authors have to state where their novelty lies and what feature they achieve others don,t show.

Author Response

Response to Reviewer 3 Comments

Point 1: If you use GO as abreviation in abstract please explain what that is? The other part which is not clear written out what is the novelty and goal of this work? please define a clear goal in abstract

Response 1: The GO in the article is an abbreviation for graphene oxide and has been noted in the article. A clear goal has been added to the abstract based on your requirements.

Point 2: A scientific paper needs to show precise results and Figures with a proper discussion. There is no proper discussion with references to other work presented here. As example if you show FTIR or Raman with signals you need to give the origin of the lines (references). This has to be included as well for other experimental data.

Response 2:References have been added to the article according to your requirements.

Point 3: There are no error bars shown in your experiemtal data. Did you investigate only one sample? Please include error bars and explain where they come from.

Response 3: In the article, only the part of the XPS test uses the tabular data. In this part, we only tested it once. This part maybe not involve data errors.

Point 4:

The addition of polymers on graphene oxide over click chemistry is well known and there many reviews and papers written about this topic

As example: 

Y. Pan, H. Bao,N.G. Sahoo, T.Wu, L. Li,Water-soluble poly(N-isopropylacrylamide)–
graphene sheets synthesized via click chemistry for drug delivery, Adv. Funct.
Mater. 21 (2011) 2754–2763.

The authors have to state where their novelty lies and what feature they achieve others don,t show.

Response 4: In our experiments, the GO-g-PNIPAAm nanomaterial was synthesized by the ATRP method. Compared to conventional polymerization methods, the ATRP process is applied in a variety of solvents and requires relatively mild reaction conditions. The polymer can be grown on the surface of materials including GO and nanoparticles, and the prepared polymer has a low molecular weight dispersibility index and an adjustable chain morphology and size, which is an effective way to carry out living polymerization.This part has been supplemented in the introduction of the article.

Round 2

Reviewer 1 Report

The paper can be published in the actual version

Author Response

no response.

Reviewer 2 Report

After revising the answer of the authors I still miss experimental data that actually confirm that the novel membranas have a positive thermo-responsive effect on the permeability and authors simply present a rutinely membrane characterization protocol. Either authors change their claiming on thermo-responsive membrane quality or they must include experimental evidences of that characteristic.

Author Response

Dear Reviewer,

We have change the title , abstract&conclusion of the article according to your suggestion.

Please check it.

Reviewer 3 Report

All issues addressed. Manuscript now acceptable

Author Response

no response.

Round 3

Reviewer 2 Report

I am satisfied with the response of authors to my previous recommendations.